# The Difference in Paraspinal Muscle Parameters and the Correlation with Health-Related Quality of Life among Healthy Individuals, Patients with Degenerative Lumbar Scoliosis and Lumbar Spinal Stenosis

**DOI:** 10.3390/jpm13101438

**Published:** 2023-09-26

**Authors:** Yinhao Liu, Lei Yuan, Yan Zeng, Jiajun Ni, Shi Yan

**Affiliations:** 1Department of Orthopedics, Peking University Third Hospital, 49 North Garden Road, Beijing 100191, China; liuyinhao1995@163.com (Y.L.); yuanleipku@163.com (L.Y.); jiajunni2019@163.com (J.N.); 1710301114@pku.edu.cn (S.Y.); 2Beijing Key Laboratory of Spinal Disease Research, Peking University Third Hospital, 49 North Garden Road, Beijing 100191, China; 3Engineering Research Center of Bone and Joint Precision Medicine, Ministry of Education, Peking University Third Hospital, 49 North Garden Road, Beijing 100191, China; 4Peking University Third Clinical College, Peking University Health Science Center, Haidian District, No. 38 Xueyuan Road, Beijing 100191, China

**Keywords:** paraspinal muscle degeneration, degenerative lumbar scoliosis, lumbar spinal stenosis, health-related quality of life (HRQOL)

## Abstract

(1) Background: Paraspinal muscle degeneration affects the quality of life in patients with degenerative lumbar scoliosis (DLS) and lumbar spinal stenosis (LSS). We aimed to describe the characteristics and differences in the paraspinal muscle parameters between patients with DLS and LSS and investigate their correlation with health-related quality of life (HRQOL). (2) Methods: There were forty-four participants in each group, namely the DLS, LSS, and healthy groups, who were matched at a ratio of 1:1 according to age, sex, and BMI. Differences in paraspinal muscle parameters among the three groups were compared using analysis of variance or the Mann–Whitney U test, and paraspinal muscle degeneration and HRQOL were analyzed using Spearman’s correlation analysis. (3) Results: In the upper lumbar, the psoas (PS), quadratus lumborum (QL), and multifidus (MF) cross-sectional area (CSA) in the DLS group were smaller than those in the other groups. In the lower lumbar region, the CSA of the PS, QL, erector spinae (ES), and gross CSA (GCSA) of the MF in the DLS group were not significantly different from those in the LSS group. These values were lower than those observed in the healthy group. The lean muscle fat index (LMFI) of the MF and ES groups was higher than those of the other groups. Regarding HRQOL, we found that PS and MF CSA were strongly associated with HRQOL in healthy individuals; however, only MF was associated with HRQOL in the LSS group. (4): Conclusion: PS in the upper lumbar region and MF degeneration were more severe in patients with DLS than in those with LSS. ES degeneration was similar between the LSS and DLS groups. MF muscle atrophy in patients with LSS and asymmetric changes in the MF in DLS are associated with quality of life.

## 1. Introduction

Degenerative lumbar scoliosis (DLS) is a degenerative lumbar disease that often leads to severe lower back and lower limb pain, which affects patients’ quality of life [1]. Previous studies have suggested that the asymmetric degeneration of the intervertebral disc and facet joint is the cause of its pathogenesis, which begins with intervertebral disc degeneration [2]. The paraspinal muscles, including the psoas (PS) major, quadratus lumborum (QL), multifidus (MF), and erector spinae (ES), are important for maintaining spinal stability and mobility, and their degeneration is often associated with several degenerative lumbar diseases [3,4,5]. Yawara et al. found that lower muscle mass may be involved in the progression of spinal deformities and increased low back pain [6]. Xie et al. found that an asymmetric paraspinal muscle is associated with an increased Cobb angle [7]. These studies suggest that paraspinal muscle degeneration is involved in the physiological processes of DLS.

Lumbar spinal stenosis (LSS) is one of the most common degenerative diseases of the lumbar spine, and paraspinal muscle degeneration is correlated with quality of life. Takahiro et al. found a correlation between MF muscle atrophy and pain intensity in patients with LSS [8]. Another study found that MF cross-sectional area (CSA) reduction and atrophy were associated with poorer LSS outcomes [9]. Although DLS and LSS share many symptoms, paraspinal muscle degeneration may vary. Yagi et al. compared the differences between the MF and PS major muscles in patients with DLS and LSS; however, these were limited to the L5–S1 level [10]. To the best of our knowledge, the present studies on DLS and LSS tend to be limited to single or partial paraspinal muscles.

Therefore, this study aimed to elaborate on the characteristics and differences in the paraspinal muscles between DLS and LSS. We also included healthy individuals as baseline values to better describe the degree of degeneration. We also investigated the association between the paraspinal muscle parameters and health-related quality of life (HRQOL) in these patients.

## 2. Materials and Methods

### 2.1. Inclusion and Exclusion Criteria

This was a prospective cross-sectional study. The DLS and LSS groups comprised patients who underwent surgery between February 2011 and March 2022. The inclusion criteria for patients in the DLS group were as follows: age > 40, Cobb angle > 10°, and the presence of apical vertebrae in the lumbar region. Patients with tumors, infections, traumatic spine pathology, revision surgery, or secondary scoliosis due to other etiologies (congenital and neuromuscular scoliosis) were excluded from the study. The inclusion criteria for patients in the LSS group were as follows: age > 40 and diagnosis of LSS. Patients with infections, fractures, neuromuscular disease, hip or knee joint disease, revision surgery, or spinal deformities were excluded. The healthy group underwent detailed history-taking before participating in the study and was prospectively recruited from March 2021 to June 2021. The inclusion criteria were as follows:Age > 40; No severe lower back pain in the past 6 months; No history of radicular symptoms; No coronal deformity or lumbar spondylolisthesis; No history of spinal scoliosis or surgery. 

All participants had complete imaging data.

### 2.2. HRQOL Measurements

HRQOL questionnaires were used to assess quality of life, including the visual analog scale (VAS) scores for the back and leg, the Oswestry Disability Index (ODI), and the Japanese Orthopedic Association (JOA) scoring system. The SF-36 questionnaire was also administered for the self-assessment of quality of life. All HRQOL values were extracted from electronic medical records. The SF-36 Physical Component Summary (PCS) scale and Mental Component Summary (MCS) scale scores were calculated based on normal data. Due to the lack of data on the mainland population, this study was conducted according to the Hong Kong version [11].

### 2.3. Imaging Evaluation

Anteroposterior and lateral standing radiographs, including those of the entire spine and pelvis, were obtained from all the subjects. Using the picture archiving and communication system (PACS) (GE Healthcare, Mount Prospect, IL, USA), the following parameters were measured: Cobb angle, coronal vertical axis from the central sacral vertical line (CSVL), thoracic kyphosis (TK, T5–T12), thoracolumbar kyphosis (TLK), lumbar lordosis (LL, L1–S1), lordosis of L4–S1, pelvic incidence (PI), pelvic tilt (PT), sacral slope (SS), sagittal vertical axis (SVA), and T1-Pelvic angle (TPA).

Magnetic resonance imaging (MRI) was performed on all participants using a 3.0 T system (Siemens, Germany or General Electric, Boston, MA, USA) to measure the muscle area and T2 signal intensity. The axial MRI images were aligned parallel to the middle of each disc at L2–L3, L3–L4, L4–L5, and L5–S1. The PS, QL, MF, and ES conditions were analyzed using the CSA and signal intensity (SI) from axial T2-weighted images. In order to reduce bias, the muscle area was divided by the intervertebral disc area at the same level and multiplied by 100 (muscle CSA/disc CSA × 100) to represent the lumbar muscularity in each individual. Similarly, the degree of fat change was estimated as the muscle fat index at each level by multiplying the muscle–subcutaneous fat SI ratio by 100. The region of interest of the lean muscle tissue area, excluding fatty infiltration, was drawn to determine the functional CSA (FCSA), defined as the lean muscle fat index (LMFI). The gross CSA (GCSA) was determined by drawing the muscle’s outer perimeter, including any intramuscular fat area. The GCSA was not measured in the PS and QL because it was too difficult to distinguish their borders, and the T2 signal intensities of the PS and QL were measured using the FCSA instead. To compare the differences in muscle area between the concave and convex sides of the apical vertebrae in patients with DLS, the CSA difference index (CDI) was adopted as the evaluation index: CDI = (FCSA_concave_/FCSA_convex_) × 100%.

### 2.4. Statistical Analysis

Data were analyzed using the SPSS software version 24 (SPSS Inc., Chicago, IL, USA). Descriptive results were expressed as mean and standard deviation (SD) for continuous variables with an approximately normal distribution. Categorical values were presented as frequencies and percentages. The normality and homogeneity of variance tests were performed for relevant data from the three groups. Analysis of variance or the Mann–Whitney U test of independent samples was used to verify whether there were differences among the three groups. The relationship between paraspinal muscle activity and HRQOL was evaluated using Spearman’s correlation coefficients. Statistical significance was set at *p* < 0.05.

## 3. Results

### 3.1. Patient Characteristics

A total of 132 participants (44 in each group) were enrolled, and each group was matched at a 1:1 ratio according to age, sex, and BMI. The mean ages of participants in the DLS, LSS, and healthy groups were 58.11 ± 7.40, 58.36 ± 7.24, and 57.84 ± 7.40, respectively. In the DLS group, 26 and 18 patients exhibited curved vertices on the left and right sides, respectively. In this population, the apical vertebrae were at L2, L2/3, L3, L3/4, and L4 in 12, 5, 16, 7, and 4 patients. The clinical characteristics and radiographic parameters of each group are shown in Table 1. 

Regarding the local spinal parameters, the healthy group had larger TK and LL values and smaller TLK values than the DLS and LSS groups. PI values were similar among the three groups; however, the DLS group had the highest PT and PI-LL values. Finally, regarding global sagittal parameters, the DLS group had the highest SVA and TPA values, followed by the LSS group.

The HRQOL score parameters for each group are listed in Table 2. 

The VAS scores for back pain, S-F36 GH, SF-36 SF, and SF-36 PCS scores differed significantly among the three groups. There were no significant differences in the other scores between the DLS and LSS groups; all were different from those of the healthy group.

### 3.2. Differences in Paraspinal Muscle Degeneration among the Three Groups

In the upper lumbar (L2–4), the PS, QL, and MF CSA in the DLS group were smaller than in the other groups. In the lower lumbar (L4–5 and L5–S1), the CSA of the PS, QL, ES, and GCSA of the MF in the DLS group were not significantly different from those in the LSS group. These values were lower than those observed in the healthy group. The MF FCSA in the DLS group was the smallest and that in the healthy group was the largest. The ES CSA did not differ from that of the LSS group, and the CSAs of both groups were smaller than that of the healthy group (Table 3).

The LMFI of the PS was significantly different among the three groups, with the smallest in the healthy group and the largest in the LSS group. In the upper lumbar region, the QL LMFI of DLS was not statistically different between the DLS and LSS groups and was larger than that of the healthy group; in contrast, at the L4–L5 level, there were no statistical differences among the three groups. Regarding the LMFI of the MF and ES, although the LSS group was smaller than the DLS group, there was no statistical difference between the two groups, and both were larger than those of the healthy group (Table 4).

### 3.3. Association between HRQOL and Paraspinal Muscle Parameters

We performed a correlation analysis between HRQOL and the mean paraspinal muscle CSA for all samples, and the results are shown in Table 5. 

The mean CSA was negatively correlated with the VAS score and ODI. The correlation was stronger for the back pain VAS score than for the leg pain VAS score. CSA was positively correlated with SF-36 scores for each item, and when compared to SF-36 MCS, SF-36 PCS showed a stronger correlation.

We also analyzed the correlation between HRQOL and mean CSA in each group. The results for the healthy group are presented in Table 6. 

PS CSA negatively correlated with VAS back scores and positively correlated with JOA, SF-36 Physical Functioning, and Bodily Pain scores. The MF FCSA was also negatively correlated with the VAS back scores and positively correlated with many sub-terms of the SF-36 questionnaire. In the LSS group, only MF FCSA was associated with HRQOL, particularly SF-36 (Table 7). 

Finally, we did not find a clear correlation between the HRQOL and mean paraspinal muscle CSA in the DLS group.

CDI and HRQOL were validated via Spearman’s analysis to investigate whether asymmetric changes in the paraspinal muscles in the DLS group affected quality of life. The results showed that MF-CDI was associated with HRQOL (Table 8).

## 4. Discussion

This study measured paraspinal muscle parameters and HRQOL in patients with DLS, LSS, and healthy individuals. We explored the characteristics of paraspinal muscles in different disease groups and the correlation between paraspinal muscles and quality of life. Previous studies have shown that age, sex, and BMI are closely associated with paraspinal muscle degeneration [12,13,14]. To increase comparability, we performed a 1:1 matching among the three groups. As shown in Table 1, there were no differences in demographic characteristics among the three groups, indicating that the three groups were well-matched.

We found that in the upper lumbar spine (L2–4), the PS degeneration was manifested as muscle atrophy in DLS group and as fat infiltration in LSS group, and the atrophy of MF is more obvious in DLS patients. In the lower lumbar spine, significant muscle atrophy and fat infiltration of MF and ES were observed in both groups. In the healthy group, PS and MF were correlated with lower back pain and SF-36 scores, but in the LSS group, only MF CSA correlated with HRQOL. We did not find a correlation between the mean muscle CSA and quality of life in the DLS group, but the asymmetry change in the MF was correlated with HRQOL. First, there were no significant differences in local parameters (TK, LL, and TLK) between the DLS and LSS groups; however, the global sagittal imbalance in DLS was greater than that in LSS, and both groups of patients were different from healthy individuals. The PT angle was the largest in the DLS group, suggesting that sagittal balance was maintained through pelvic retroversion. Hasegawa et al. compared patients with DLS and LSS and obtained similar results. However, they found that LL also differed between the two groups, which may have resulted from its lack of matching for age [15]. Farrokhi et al. compared 48 age- and sex-matched patients with LSS with healthy volunteers and obtained results consistent with our research [16]. Compared to healthy individuals, patients with DLS and LSS have an obvious sagittal imbalance, and the overall sagittal imbalance in patients with DLS is more severe than that in patients with LSS.

MRI is the gold standard for estimating muscle mass among the numerous measurement techniques. The degree of muscle atrophy was evaluated by measuring the CSA, and the degree of fat infiltration was evaluated using the MRI signal ratio. We found that in the upper lumbar spine (L2–4), PS atrophy was the most severe in the DLS group; however, there was no significant difference between patients with LSS and healthy individuals. However, fat infiltration in the PS was more severe in the LSS group than in the DLS group, and the PS major muscle mainly plays a role in hip flexion and spinal stability [17]. Since there is little fat infiltration, muscle degeneration is often consistent with muscle atrophy. Although the fat infiltration (FI) of the PS was more severe in the LSS group, its CSA in the upper lumbar spine was still larger than that in the DLS group. This may be because the PS muscle originates from the T12 vertebra, lumbar vertebra, and side of the intervertebral disc of L1–5, ending at the lesser trochanter of the femur. With decreased LL and PT, the PS muscle in patients with DLS became shorter than in patients with LSS. Therefore, the density of muscle bundles per unit area increased, resulting in a higher signal intensity than in the LSS group. This may also explain the difference in QL degeneration between the two groups.

Fortin et al. found that the volumes of the MF and ES atrophy with age and that degeneration occurs from the lower lumbar spine to the upper side [18]. In the upper lumbar region, the MF of patients with LSS did not show significant atrophy; in contrast, the MF atrophy of patients with DLS was significant. In the lower lumbar region, although the GCSA of the LSS group was not different from that of the DLS group, its FCSA was still larger, indicating that the degeneration of the MF in the upper lumbar spine in the LSS group was dominated by fat infiltration; in contrast, the degeneration of the MF in the lower lumbar spine was accompanied by fat infiltration and muscle atrophy. These two changes were observed in the DLS group, and the lower lumbar muscle atrophy was more severe than that in the LSS group. The MF muscle is thought to be innervated by a nerve root (the medial branch of the dorsal ramus), which extends caudally to the 3–4 vertebral levels, and denervation is associated with the increased fatty infiltration of muscles [19]. The degeneration of the lumbar spine starts from the lower lumbar spine, and with denervation, fat infiltration occurs in both groups; however, in patients with DLS, the paraspinal muscles undergo biomechanical changes with the development of scoliosis. Biomechanical studies have shown that the load of the MF muscle increases by 1.5 kg/cm^2^ for every 1 cm deviation from the midline of the apex of the vertebral body on the convex side [20]. The lower lumbar spine bears more pressure; therefore, the degeneration rate of the paraspinal muscles of the lower lumbar spine is more obvious than that of the upper lumbar spine. Wang et al. [21] found that paraspinal muscle degeneration was more severe in patients with lumbar spinal stenosis than in healthy individuals and was more common in the MF muscles. Yagi et al. [10] found a significantly smaller CSA in both the MF and PS in the DLS group than in the matched LSS cohort, but only at L5–S1, similar to the results of our research.

Regarding the ES, patients with DLS and LSS showed obvious muscle atrophy and fatty infiltration compared to patients in the healthy group. DLS and LSS can cause intermittent claudication; in contrast, the flexion of the lumbar spine can increase the area of the intervertebral foramina and reduce pain. However, this abnormal position can lead to ES apraxia. At the same time, patients with DLS are more likely to experience sagittal imbalance, further aggravating atrophy. This also explains why the CSA was smaller and the LMFI was larger in the DLS group than in the LSS group, although there was no significant difference between the two groups.

Many studies have confirmed the correlation between paraspinal muscle degeneration and quality of life; however, the evaluation methods differ. Miki et al. [8] used the NRS, RMDQ, and EQ-5D scales and found that a decreased MF CSA was associated with lower back pain. Getzmann, J.M., et al. [22] found that fatty muscle infiltration was associated with higher rates of disability and worse HRQOL in patients with EQ-5D-3L scores. Han et al. [23] found a significant correlation among paraspinal muscle endurance, VAS scores, and ODI scores in patients with LSS. To reduce the differences caused by the different reliabilities of the scale, we used more common evaluation methods, such as the VAS score, ODI, and JOA as standards. The SRS-22 is a commonly used questionnaire for assessing the quality of life in patients with scoliosis [24]. However, considering the composition of the study population, we used a more generalizable questionnaire like the SF-36.

We conducted a correlation analysis of all subject’s mean CSA and HRQOL. Paraspinal muscle CSA negatively correlated with VAS score and ODI, positively correlated with JOA and SF-36 scores, and more strongly correlated with SF-36 PCS than SF-36 MCS scores. However, considering the different pathological characteristics of the three groups, we conducted a stratified analysis of the three groups. Interestingly, the correlation between the paraspinal muscles and HRQOL was not the same across the groups. In the healthy group, PS and MF were correlated with lower back pain and SF-36 scores. In the LSS group, only MF correlated with HRQOL. However, there was no correlation between the mean CSA and quality of life scores in the DLS group. We found that the degree of MF asymmetry in the DLS group correlated with a higher ODI and lower SF-36 RF, BP, GH, and PCS scores, consistent with previous studies’ results [25,26].

The stabilization system of the spine comprises the vertebral body, intervertebral discs, ligaments, and paraspinal muscles [27]. The degeneration of these structures leads to clinical symptoms and affects the quality of life. In healthy individuals, the degeneration of the spinal stability system mainly occurs in the paraspinal muscles; therefore, there is a good correlation between the paraspinal muscle CSA and HRQOL score. In lumbar spinal stenosis, in addition to the degeneration of paraspinal muscles, structures such as the intervertebral discs and facet joints also undergo degeneration. Therefore, as the most important stable muscle group, only the MF correlated with quality of life, consistent with previously reported results [8,28,29]. DLS is a type of three-dimensional structural change of the vertebral body. Although the degeneration of paraspinal muscles is more obvious in patients with DLS, the overall structural changes (such as sagittal and coronal imbalance) and other factors will more significantly affect the quality of life of patients [30,31,32,33,34]. These factors reduced the correlation between paraspinal muscle atrophy and HRQOL. Compared with the atrophy of the paraspinal muscles, the asymmetric change in the MF in DLS affects quality of life more clearly. Tang et al. found similar results [35]. Tsutsui et al. found that the degree of apical rotation of the vertebral body before surgery was significantly correlated with postoperative residual lower back pain [36]. This also illustrates that severe vertebral deformity and asymmetric paraspinal muscles strongly correlate with HRQOL.

This study had some limitations. First, as this was a cross-sectional study, we could not prove a causal relationship between paraspinal muscle degeneration and disease. Secondly, although strict matching was performed, the sample size among the groups was relatively small. Third, we only examined the correlation between the paraspinal muscles and HRQOL and did not examine other factors, such as sagittal parameters and disc degeneration. Therefore, prospective studies with larger sample sizes are needed to investigate the role of the paraspinal muscles in different lumbar spine diseases.

## 5. Conclusions

Compared with healthy individuals, the degeneration of PS and MF in the upper lumbar region was mainly due to fatty infiltration in patients with LSS, and both fatty infiltration and muscle atrophy were present in the lower lumbar region. In patients with DLS, the paraspinal muscles showed two changes. The PS in the upper lumbar region and MF degeneration were more severe in patients with DLS than those with LSS. ES degeneration was similar between the LSS and DLS groups. MF muscle atrophy in patients with LSS and asymmetric changes in the MF in DLS are associated with quality of life. We speculated that the more obvious the degeneration of the spinal structure, the lower the correlation between the paraspinal muscles and quality of life.

## Figures and Tables

**Table 1 jpm-13-01438-t001:** Clinical characteristics and radiographic parameters of whole spine X-ray in three groups.

Parameters	Scoliosis	Stenosis	Normal	*p*
Age (years)	58.11 ± 7.40	58.36 ± 7.24	57.84 ± 7.40	0.946
Sex (male/female)	11/33	11/33	11/33	/
BMI (kg/m^2^)	24.01 ± 1.74	23.69 ± 1.98	23.47 ± 1.92	0.402
Cobb (°)	25.56 ± 7.88 ^†‡^	4.89 ± 3.69 ^‡^	5.59 ± 2.15 ^†^	<0.000
CVA (mm)	18.56 ± 14.37 ^†^	11.91 ± 10.24	8.70 ± 6.42 ^†^	<0.000
TK (°)	15.97 ± 16.31 ^‡^	8.13 ± 12.76 ^†^	30.39 ± 10.48 ^†‡^	0.007
TLK (°)	19.40 ± 20.48 ^‡^	23.81 ± 11.06 ^†^	6.87 ± 9.34 ^†‡^	<0.000
LL (°)	14.88 ± 21.89 ^‡^	24.78 ± 16.54 ^†^	36.92 ± 10.17 ^†‡^	<0.000
L4-S1(°)	32.50 ± 16.53	28.88 ± 11.86 ^†^	38.09 ± 6.71 ^†^	0.001
SS (°)	24.08 ± 10.55	32.23 ± 9.80	36.27 ± 7.53	<0.000 *
PT (°)	23.88 ± 10.55	17.49 ± 7.51	11.99 ± 5.84	<0.000 *
PI (°)	47.92 ± 11.27	49.71 ± 10.67	48.23 ± 9.54	0.696
PI-LL (°)	33.04 ± 20.52	24.92 ± 12.84	11.31 ± 10.59	<0.000 *
SVA (mm)	41.29 ± 49.99	24.85 ± 36.87	3.05 ± 14.81	<0.000 *
TPA (°)	20.98 ± 10.79	14.56 ± 7.26	7.44 ± 4.88	<0.000 *

“†” and “‡” represent the two groups that had significant differences, and “*” represents the three groups that had significant differences with each other (all adjusted *p* < 0.05).

**Table 2 jpm-13-01438-t002:** Health-related quality of life score (HRQOL) parameters in three groups (*: *p* < 0.05).

Parameters	Scoliosis	Stenosis	Normal	*p*
VAS back	6.16 ± 1.94	5.16 ± 2.23	2.64 ± 1.22	<0.000 *
VAS leg	4.98 ± 2.73 ^†^	5.43 ± 2.43 ^‡^	2.27 ± 1.70 ^†‡^	<0.000
ODI	50.36 ± 16.64 ^†^	46.23 ± 17.31 ^‡^	4.45 ± 4.38 ^†‡^	<0.000
JOA	15.80 ± 4.38 ^†^	15.55 ± 5.36 ^‡^	26.61 ± 2.47 ^†‡^	<0.000
SF-36 PF (Physical Functioning)	42.25 ± 23.77 ^†^	46.36 ± 21.76 ^‡^	89.09 ± 10.80 ^†‡^	<0.000
SF-36 RP (Role—Physical)	9.10 ± 25.34 ^†^	15.34 ± 34.61 ^‡^	91.48 ± 24.08 ^†‡^	<0.000
SF-36 BP (Bodily Pain)	39.59 ± 22.30 ^†^	44.75 ± 28.51 ^‡^	76.07 ± 19.83 ^†‡^	<0.000
SF-36 GH (General Health)	48.09 ± 26.63	59.29 ± 24.17	69.96 ± 17.24	<0.000 *
SF-36 VT (Vitality)	51.93 ± 24.69 ^†^	61.36 ± 24.83 ^‡^	73.52 ± 17.10 ^†‡^	<0.000
SF-36 SF (Social Functioning)	44.32 ± 27.29	57.56 ± 27.59	89.21 ± 16.31	<0.000 *
SF-36 RE (Role—Emotional)	20.41 ± 38.99 ^†^	17.41 ± 36.99 ^‡^	88.07 ± 23.18 ^†‡^	<0.000
SF-36 MH (Mental Health)	59.64 ± 23.10 ^†^	62.73 ± 23.38 ^‡^	82.82 ± 11.23 ^†‡^	<0.000
SF-36 PCS	44.27 ± 5.35	47.03 ± 6.48	58.70 ± 4.78	<0.000 *
SF-36 MCS	46.06 ± 8.55 ^†^	47.79 ± 8.87 ^‡^	56.15 ± 5.23 ^†‡^	<0.000

“†” and “‡” represent the two groups that had significant differences (all adjusted *p* < 0.05).

**Table 3 jpm-13-01438-t003:** Lumbar muscularity (CSA of muscle–disc ratio×100) of the paraspinal muscles of the three groups using MRI.

		Scoliosis	Stenosis	Healthy	*p*
PS CSA	L2–3	25.39 ± 7.92 ^†‡^	31.41 ± 14.73 ^‡^	34.61 ± 11.65 ^†^	<0.001
	L3–4	33.27 ± 10.63	40.80 ± 13.33	48.06 ± 14.20	<0.001 *
	L4–5	44.27 ± 13.32 ^‡^	49.67 ± 15.34 ^†^	62.11 ± 17.99 ^†‡^	<0.001
	L5–S1	48.35 ± 15.92^‡^	55.39 ± 17.42 ^†^	69.15 ± 19.48 ^†‡^	<0.001
QL CSA	L2–3	12.16 ± 5.38 ^†‡^	16.70 ± 8.38 ^‡^	16.24 ± 5.63 ^†^	0.002
	L3–4	16.53 ± 5.36 ^†‡^	20.71 ± 6.22 ^‡^	20.78 ± 6.39 ^†^	0.001
	L4–5	15.66 ± 7.87 ^‡^	17.12 ± 8.02 ^†^	22.95 ± 7.81 ^†‡^	<0.001
	L5–S1	/	/	/	/
GCSA of MF	L2–3	19.52 ± 8.65 ^†‡^	27.31 ± 16.86 ^‡^	27.67 ± 8.33 ^†^	0.002
	L3–4	25.64 ± 9.99 ^†‡^	33.81 ± 13.05 ^‡^	36.12 ± 7.64 ^†^	<0.001
	L4–5	37.38 ± 16.67 ^‡^	41.83 ± 10.37 ^†^	52.38 ± 10.50 ^†‡^	<0.001
	L5–S1	53.22 ± 20.43 ^‡^	54.68 ± 16.82 ^†^	71.66 ± 16.37 ^†‡^	<0.001
FCSA of MF	L2–3	11.22 ± 4.84 ^†‡^	18.94 ± 14.43 ^‡^	17.97 ± 5.69 ^†^	<0.001
	L3–4	14.82 ± 5.98 ^†‡^	23.72 ± 12.07 ^‡^	23.61 ± 5.28 ^†^	<0.001
	L4–5	21.54 ± 10.52	26.89 ± 7.77	33.29 ± 6.97	<0.001 *
	L5–S1	27.71 ± 12.08	34.26 ± 13.29	43.07 ± 12.13	<0.001 *
GCSA of ES	L2–3	93.52 ± 19.87 ^‡^	99.05 ± 25.57 ^†^	120.49 ± 20.54 ^†‡^	<0.001
	L3–4	80.15 ± 18.49 ^‡^	83.15 ± 21.10 ^†^	106.47 ± 21.08 ^†‡^	<0.001
	L4–5	72.53 ± 17.95 ^‡^	73.00 ± 20.35 ^†^	95.41 ± 20.42 ^†‡^	<0.001
	L5–S1	48.83 ± 22.73 ^‡^	45.43 ± 19.07 ^†^	70.91 ± 27.21 ^†‡^	<0.001
FCSA of ES	L2–3	69.79 ± 16.99	79.00 ± 21.51	97.39 ± 18.87	<0.001 *
	L3–4	55.68 ± 17.15 ^‡^	61.39 ± 17.13 ^†^	80.37 ± 18.53 ^†‡^	<0.001
	L4–5	45.08 ± 13.88 ^‡^	46.84 ± 16.36 ^†^	63.97 ± 16.25 ^†‡^	<0.001
	L5–S1	26.03 ± 15.87 ^‡^	23.63 ± 14.83 ^†^	37.21 ± 17.58 ^†‡^	<0.001

“†” and “‡” represent the two groups that had significant differences, and “*” represents the three groups that had significant differences with each other (all adjusted *p* < 0.05).

**Table 4 jpm-13-01438-t004:** Degree of fatty change (mean signal intensity of muscle–subcutaneous fat ratio×100) of the paraspinal muscles of three groups using MRI.

		Scoliosis	Stenosis	Healthy	*p*
LMFI of PS	L2–3	12.22 ± 5.78	14.31 ± 4.33	9.82 ± 2.34	<0.001 *
	L3–4	12.51 ± 5.51	14.60 ± 4.47	9.33 ± 2.78	<0.001 *
	L4–5	11.83 ± 5.01	13.59 ± 4.16	8.85 ± 3.07	<0.001 *
	L5–S1	11.26 ± 5.33	13.32 ± 5.34	8.97 ± 2.39	<0.001 *
LMFI of QL	L2–3	14.83 ± 7.21 ^†^	15.74 ± 5.31 ^‡^	11.66 ± 2.94 ^†‡^	0.002
	L3–4	15.03 ± 7.54 ^†^	15.23 ± 5.40 ^‡^	12.36 ± 3.20 ^†‡^	0.032
	L4–5	16.40 ± 8.46	18.55 ± 7.28	16.86 ± 6.76	0.377
	L5–S1	/	/	/	/
LMFI of MF	L2–3	22.55 ± 7.73 ^†^	19.73 ± 3.81 ^‡^	14.23 ± 5.20 ^†‡^	<0.001
	L3–4	23.23 ± 8.63 ^†^	19.81 ± 4.28 ^‡^	13.86 ± 5.03 ^†‡^	<0.001
	L4–5	23.58 ± 8.35 ^†^	21.46 ± 7.20 ^‡^	14.34 ± 4.28 ^†‡^	<0.001
	L5–S1	24.53 ± 9.73 ^†^	21.38 ± 6.16 ^‡^	14.90 ± 4.13 ^†‡^	<0.001
LMFI of ES	L2–3	19.52 ± 6.04 ^†^	17.45 ± 3.55 ^‡^	13.67 ± 4.28 ^†‡^	<0.001
	L3–4	21.24 ± 7.09 ^†^	17.95 ± 4.04 ^‡^	14.68 ± 4.01 ^†‡^	<0.001
	L4–5	22.19 ± 8.29 ^†^	19.47 ± 5.60 ^‡^	17.01 ± 4.50 ^†‡^	<0.001
	L5–S1	26.66 ± 7.69 ^†^	24.67 ± 6.93 ^‡^	20.53 ± 5.16 ^†‡^	<0.001

“†” and “‡” represent the two groups that had significant differences, and “*” represents the three groups that had significant differences with each other (all adjusted *p* < 0.05).

**Table 5 jpm-13-01438-t005:** Correlation analysis between HRQOL and mean paraspinal muscle CSA for all samples (*: *p* < 0.05, **: *p* < 0.01).

Parameters	PS	QL	FCSA of MF	FCSA of ES
VAS Back	−0.375 **	−0.311 **	−0.413 **	−0.379 **
VAS Leg	−0.337 **	−0.217 **	−0.266 **	−0.339 **
JOA	0.358 **	0.296 **	0.417 **	0.499 **
ODI	−0.385 **	−0.352 **	−0.460 **	−0.515 **
SF-36 PF	0.387 **	0.292 **	0.437 **	0.438 **
SF-36 RP	0.396 **	0.384 **	0.426 **	0.470 **
SF-36 BP	0.352 **	0.371 **	0.395 **	0.404 **
SF-36 GH	0.214 *	0.272 **	0.351 **	0.188 *
SF-36 VT	0.215 *	0.238 **	0.372 **	0.225 **
SF-36 SF	0.277 **	0.334 **	0.406 **	0.420 **
SF-36 RE	0.344 **	0.259 **	0.368 **	0.361 **
SF-36 MH	0.245 **	0.225 **	0.325 **	0.284 **
SF-36 PCS	0.421 **	0.417 **	0.494 **	0.475 **
SF-36 MCS	0.246 **	0.240 **	0.358 **	0.293 **

**Table 6 jpm-13-01438-t006:** Correlation analysis between HRQOL and mean paraspinal muscle CSA in healthy individuals. (*: *p* < 0.05, **: *p* < 0.01).

Parameters	PS	QL	FCSA of MF	FCSA of ES
VAS Back	−0.353 * (*p* = 0.019)	−0.281	−0.328 * (*p* = 0.030)	−0.186
VAS Leg	−0.250	−0.102	−0.201	−0.303 * (*p* = 0.046)
JOA	0.370 * (*p* = 0.013)	0.211	0.275	0.283
ODI	−0.253	−0.275	−0.139	−0.238
SF-36 PF	0.375 * (*p* = 0.012)	0.240	0.413 ** (*p* = 0.005)	0.420 ** (*p* = 0.004)
SF-36 RP	0.080	0.210	0.399 ** (*p* = 0.007)	0.126
SF-36 BP	0.327 * (*p* = 0.030)	0.290	0.248	0.159
SF-36 GH	0.153	0.047	0.218	0.021
SF-36 VT	0.268	0.224	0.311 * (*p* = 0.040)	0.222
SF-36 SF	0.005	0.147	0.003	0.032
SF-36 RE	0.193	0.284	0.306 * (*p* = 0.044)	0.220
SF-36 MH	0.187	0.166	−0.038	0.098
SF-36 PCS	0.292	0.223	0.444 ** (*p* = 0.003)	0.211
SF-36 MCS	0.137	0.203	0.056	0.110

**Table 7 jpm-13-01438-t007:** Correlation analysis between HRQOL and mean MF FCSA in patients with LSS.

	ODI	SF-36 RP	SF-36 BP	SF-36 VT	SF-36 SF	SF-36 RE	SF-36 MH	SF-36 PCS	SF-36 MCS
MF FCSA	−0.328 (*p* = 0.041)	0.321 (*p* = 0.034)	0.300 (*p* = 0.048)	0.329 (*p* = 0.029)	0.308 (*p* = 0.042)	0.341 (*p* = 0.023)	0.321 (*p* = 0.034)	0.347 (*p* = 0.021)	0.317 (*p* = 0.013)

**Table 8 jpm-13-01438-t008:** Correlation analysis between HRQOL and MF CDI in patients with DLS.

	ODI	SF-36 BP	SF-36 GH	SF-36 MH	SF-36 PCS
MF CDI	−0.317 (*p* = 0.036)	0.300 (*p* = 0.047)	0.422 (*p* = 0.004)	0.377 (*p* = 0.012)	0.347 (*p* = 0.018)

## Data Availability

The data presented in this study are available on request from the corresponding author.

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
