# Peer review of "The Difference in Paraspinal Muscle Parameters and the Correlation with Health-Related Quality of Life among Healthy Individuals, Patients with Degenerative Lumbar Scoliosis and Lumbar Spinal Stenosis"

_jpm, 2023, doi:10.3390/jpm13101438_

Round 1

Reviewer 1 Report

Thank you for the opportunity to review the manuscript entitled The difference in paraspinal muscle parameters and the correlation with health-related quality of life among healthy individuals, patients with degenerative lumbar scoliosis and lumbar spinal stenosis. The authors clearly did a lot of work, and appear to have designed and executed the study carefully. The results provide the basis for further study on the biomechanics of the lumbar spine in health and disease. The clinical implications of this work are not immediately obvious, but, to their credit, the authors do not make any excessive claims. At the sentence and paragraph level, the English is quite comprehensible, and the entire narrative proceeds in a sensible fashion. I am not qualified to comment on the statistical analysis, and will leave this consideration to other reviewers. Otherwise, I have no concerns about the publication of this article.

Author Response

We are very grateful to Reviewer for the affirmation and constructive comments on this study. For the study of paraspinal muscles, it is difficult to translate them into clinical implications. We hope that we can do further research in this aspect in the next study.

Reviewer 2 Report

Excellent idea trying to correlate muscle atrophy and specific muscle groups to clinical parameters, most notably HRQoL.  Begs the question as to whether specifically rehabbing certain muscle groups would reduce pain and improve function.

I found the result section difficult to follow:

   a. the tables are extensive and a bit difficult to follow, esp with all the HRQoL parameters listed - is that amount of detail really necessary and relevant (eg. table 5)? I would condense or simplify.

   b. the text between tables is not clear as to whether it is paprt of the table or the manuscript.

What is meant by line 217 (it does not make sense )?

Line 256 discusses this theory concerning ES muscle atrophy and FI between DLS and LSS; however, there were no significant differences between the two groups so why the elaborate speculation?

A point worth more clearly emphasizing was the finding that while muscle atrophy was significant in patients with DLS, it was difficult to separate the effects of their spinal imbalance from their muscle problem with regard to HRQoL scores.  Trying to micro-analyze the different muscle groups atrophy/fat infiltration and their potential significance seems moot.

Overall, the reader might benefit from a clearer discussion of the muscle groups that were significantly abnormal, as they related to the type of spinal disorder, and how that correlated with disability.  I would separate these results from the speculation as to the cause, discussing that in the discussion section.

Author Response

Thank you for your summary. We really appreciate your efforts in reviewing our manuscript. We have revised the manuscript accordingly. Our point-by-point responses are detailed below.

Comments 1: The tables are extensive and a bit difficult to follow, esp with all the HRQOL parameters listed - is that amount of detail really necessary and relevant (eg. table 5)? I would condense or simplify.

Response 1: For Table 5, we first analyzed all the samples and found that different muscles seemed to have a good correlation with HQROL scores, but when we stratified the analysis by disease characteristics, we got different conclusions. We tried to show all statistically different for readers, so we felt that the contents of the table could not be simplify. We did not list all HRQOL parameters for the group with poor correlation between paraspinal muscles and HRQOL (eg. table 7/8).

Comments 2: The text between tables is not clear as to whether it is part of the table or the manuscript.

Response 2: We are sorry for not showing the information and have added details in the Results (Page 5, paragraph 2, and line 1).

Comments 3: What is meant by line 217 (it does not make sense)?

Response 3: We thank for the Reviewer’s reminder. What we're trying to say is “Since there is little fat infiltration, muscle degeneration is often consistent with muscle atrophy.” We have revised it in the manuscript (Page 8, paragraph 1, and line 8).

Comments 4: Line 256 discusses this theory concerning ES muscle atrophy and FI between DLS and LSS; however, there were no significant differences between the two groups so why the elaborate speculation?

Response 4: We are sorry for confusing the Reviewer. For DLS and LSS, although there is no statistical difference between the two groups, the ES CSA of DLS is smaller than LCS and the LMFI is larger than LCS group. We may have used the wrong conjoint to make readers misunderstand, and we have been corrected in the manuscript (Page 8, paragraph 3, and line 7).

Comments 5: A point worth more clearly emphasizing was the finding that while muscle atrophy was significant in patients with DLS, it was difficult to separate the effects of their spinal imbalance from their muscle problem with regard to HRQOL scores. Trying to micro-analyze the different muscle groups atrophy/fat infiltration and their potential significance seems moot.

Response 5: We appreciate the Reviewer with the professional question. As you said, spinal imbalance has an impact on the quality of life of patients, and it is difficult to exclude the influence for this study. However, according to the SRS-Schwab classification, there were few patients with significant sagittal imbalance (SVA > 9.5cm) in our study, which also minimized the bias of the study. We tried to understand the role of muscle in disability by analyzing the correlation between the different muscle degeneration and quality of life in different diseases.

Comments 6: Overall, the reader might benefit from a clearer discussion of the muscle groups that were significantly abnormal, as they related to the type of spinal disorder, and how that correlated with disability.  I would separate these results from the speculation as to the cause, discussing that in the discussion section.

Response 6: We thank for the Reviewer’s reminder and revised as suggested (Page 7, and paragraph 5).

Reviewer 3 Report

After reviewing the manuscript with the title The differences in paraspinal muscle parameters and the correlation with health-related quality of life among healthy individuals, patients with degenerative lumbar scoliosis and lumbar spinal stenosis, there are several notes:

1. The title and limitations of the study are not appropriate. The title says the patient has degenerative lumbar scoliasis and lumbar spinal stenosis. However, due to limitations, the author could not prove a causal relationship between paraspinal muscle degeneration and disease.

2. Figures 6 and 7 have been corrected to make it easier for readers to observe and understand them. Each group is separated by a horizontal line.

In general, the manuscript is well written and the data is presented adequately.

Author Response

Thank you for your precious comments and advice. We have studied the comments and queries carefully and have made corrections. The main corrections in the paper and the responses to the reviewer’s queries are as flowing:

Comments 1:The title and limitations of the study are not appropriate. The title says the patient has degenerative lumbar scoliasis and lumbar spinal stenosis. However, due to limitations, the author could not prove a causal relationship between paraspinal muscle degeneration and disease.

Response 1: The title of this research is “The difference in paraspinal muscle parameters and the correlation with health-related quality of life among healthy individuals, patients with degenerative lumbar scoliosis and lumbar spinal stenosis”, which means that we want to discuss the difference between paraspinal muscles degeneration and their correlation with quality of life in different diseases, rather than prove a causal relationship between paraspinal muscle degeneration and disease. In fact, it is difficult to prove the causal relationship between muscle degeneration and disease, we hope that we can do further research in this aspect in the next study.

Comments 2:Figures 6 and 7 have been corrected to make it easier for readers to observe and understand them. Each group is separated by a horizontal line.

Response 2: We thank the Reviewer and revised Figure 6 and 7 as suggested.